

# An emulator approach to stratocumulus susceptibility

Franziska Glassmeier[1,4], Fabian Hoffmann[1,2], Jill S. Johnson[3], Takanobu Yamaguchi[1,2], Ken S. Carslaw[3], and Graham Feingold[1]

[1]NOAA Earth Systems Research Laboratory, Chemical Sciences Division, 325 Broadway, Boulder, CO 80302, USA
[2]Cooperative Institute for Research in Environmental Sciences, University of Colorado Boulder, Boulder, CO 80309, USA
[3]University of Leeds, School of Earth and Environment, Woodhouse Lane Leeds, LS2 9JT, UK
[4]Wageningen University, Department of Environmental Sciences, Droevendaalsesteeg 4, 6708PB Wageningen, NL

**Correspondence:** Franziska Glassmeier (franziska.glassmeier@noaa.gov)

**Abstract.** The climatic relevance of aerosol-cloud interactions depends on the sensitivity of the radiative effect of clouds to cloud droplet number $N$ and liquid water path LWP. We derive the dependence of cloud fraction CF, cloud albedo $A_C$ and the relative cloud radiative effect $\text{rCRE} = \text{CF} \cdot A_C$ on $N$ and LWP from 159 large-eddy simulations of nocturnal stratocumulus. These simulations vary in their initial conditions for temperature, moisture, boundary-layer height and aerosol concentration but share boundary conditions for surface fluxes and subsidence. Our approach is based on Gaussian process emulation, a statistical technique related to machine learning. We succeed in building emulators that accurately predict simulated values of CF, $A_C$ and rCRE for given values of $N$ and LWP. Emulator-derived susceptibilities $\partial \ln \text{rCRE}/\partial \ln N$ and $\partial \ln \text{rCRE}/\partial \ln \text{LWP}$ cover the non-drizzling, fully-overcast regime as well as the drizzling regime with broken cloud cover. Theoretical results, which are limited to the non-drizzling regime, are reproduced. The susceptibility $\partial \ln \text{rCRE}/\partial \ln N$ captures the strong sensitivity of the cloud radiative effect to cloud fraction, while the susceptibility $\partial \ln \text{rCRE}/\partial \ln \text{LWP}$ describes the influence of cloud amount on cloud albedo irrespective of cloud fraction. Our emulation-based approach provides a powerful tool for summarizing complex data in a simple framework that captures the sensitivities of cloud field properties over a wide range of states.

## 1 Introduction

Aerosol perturbations can lead to changes in cloud brightness and amount via the influence of aerosol on cloud formation and various aerosol-cloud interaction (ACI) processes. Our process understanding of ACI has improved greatly over recent decades, however the radiative forcing due to ACI, especially from shallow, warm clouds, continues to dominate the uncertainty margin of the total anthropogenic forcing of the climate system (Myhre et al., 2013; Mülmenstädt and Feingold, 2018).

ACIs are notoriously hard to quantify because they pose a multi-scale problem, not only in terms of spatial and temporal scales but also in terms of the effective degrees of freedom used to describe ACI in different settings. The multi-scale spectrum of approaches to the ACI problem has been described to range from "Darwinian" (low-level, high-dimensional, complex, *reductionist*) to "Newtonian" (high-level, low-dimensional, effective, *emergent*) descriptions (Harte, 2002; Feingold et al., 2016; Mülmenstädt and Feingold, 2018).



We illustrate this notion by discussing the *relative cloud radiative effect* (rCRE) as quantified by

$$\text{rCRE} = \frac{F_{\text{clr}} - F_{\text{all}}}{F_{\text{clr}}} \approx A_{\text{C}} \cdot \text{CF}, \tag{1}$$

where $F$ denotes downwelling SW radiative fluxes at the surface under clear-sky (index clr) and all-sky (index all) conditions, and $A_{\text{C}}$ and CF denote cloud albedo and cloud fraction, respectively (Xie and Liu, 2013). By describing their effect on the radiation budget, rCRE captures the effect of clouds on climate. In the spirit of Platnick and Twomey (1994), ACI can be quantified based on the *susceptibility*, or normalized sensitivity, of the rCRE to the cloud droplet number concentration $N$:

$$\frac{\mathrm{d}\ln\text{rCRE}}{\mathrm{d}\ln N} = \frac{\partial\ln\text{rCRE}}{\partial\ln N} + \frac{\partial\ln\text{rCRE}}{\partial\ln\text{LWP}}\frac{\mathrm{d}\ln\text{LWP}}{\mathrm{d}\ln N}. \tag{2}$$

The decomposition on the right-hand side (RHS) is motivated by distinguishing cloud micro-structure as captured by $N$ from macro-structure as represented by the liquid water path, LWP. Note that this decomposition does not necessarily align with the contributions of $A_{\text{C}}$ and CF to rCRE. Aerosol effects on cloud micro-structure are associated with cloud brightness (Twomey, 1977, 1974; Boers and Mitchell, 1994; Feingold et al., 1997; Christensen and Stephens, 2011; McGibbon and Bretherton, 2017), while aerosol effects on the macro-structure of the cloud correspond to cloud amount (Albrecht, 1989; Matheson et al., 2005; Kaufman et al., 2005; Small et al., 2009; Stevens and Feingold, 2009; Zheng et al., 2010; Christensen and Stephens, 2011; Chen et al., 2014; Feingold et al., 2015; McGibbon and Bretherton, 2017).

The reductionist approach to deriving rCRE($N$,LWP) and the partial derivatives in Equation 2 starts by asking how $A_{\text{C}}(\tau)$ depends on cloud optical thickness $\tau$, followed by deriving the dependence of $\tau(\text{LWP}, N)$ on LWP and $N$, which are, in turn, functions of the meteorological and aerosol conditions. Even more complex chains of dependencies can be derived for CF($N$,LWP) and LWP($N$) because these relationships are shaped by the entire cloud field. The advantage of this approach is that each link can in principle be deduced from detailed process understanding. The disadvantage is that a large number of variables and processes need to be quantified.

The emergence-based alternative is to subsume process complexity in low-dimensional relationships that effectively describe rCRE as a function of a small set of controlling variables. In other words, this means searching for a simple relationship rCRE(LWP, $N$). Or it may mean abandoning the idea of disentangling aerosol effects on cloud microstructure ($N$) and macrostructure (LWP) altogether – similar to the definition of effective radiative forcing in Myhre et al. (2013) – and quantifying d ln rCRE/d ln N directly. While the direct nature of this approach is an obvious advantage, the challenge of the emergence-based approach lies in its data-mining, exploratory nature, and lack of a priori guidance for finding and justifying emergent relationships.

In this paper, we aim to combine the reductionist and emergence-based approaches to determine the partial derivatives in Equation 2; the LWP adjustment $\mathrm{d}\ln\text{LWP}/\mathrm{d}\ln N$ will be the topic of an upcoming paper. We demonstrate how a statistical method related to machine learning can be applied to derive system-wide relationships from the detailed process representation that is ingrained in a set of model simulations. Our contribution addresses an increasing interest in machine learning approaches within the atmospheric sciences, especially in the context of parameterizing shallow clouds (Krasnopolsky et al., 2013; Schneider et al., 2017; Brenowitz and Bretherton, 2018; Gentine et al., 2018; O'Gorman and Dwyer, 2018). This interest



in utilizing modern computational-statistical methods illustrates a community need to address a certain mismatch between traditional process-based cloud research and synthesizing approaches, especially for representing clouds in climate models.

We specifically apply the tool of *Gaussian process emulation* (Rasmussen and Williams, 2006; O'Hagan, 2006). Emulation is an established method used to extract multi-dimensional relationships from sparse data. It can be considered a form of kernel-

based supervised machine learning. In the atmospheric sciences, emulation has so far been used to investigate the relationship between model response and uncertain parameters associated with physical parameterizations and to a lesser extent boundary conditions, e. g., Lee et al. (2011), Lee et al. (2013), Johnson et al. (2015) and Posselt et al. (2016). We adapt this method to quantitatively derive relationships between cloud field properties, namely $\text{rCRE}(\text{LWP}, N)$, $A_{\text{C}}(\text{LWP}, N)$ and $\text{CF}(\text{LWP, N})$, that evolve over the course of the simulation, from simulation data.

We present state-of-the-art large-eddy simulations (LES) of stratocumulus (Section 2) and demonstrate that our approach (Section 3) successfully translates process understanding captured by the LES into a quantification of the rCRE, $A_{\text{C}}$ and CF and their relationships to LWP and N (Section 4). As an application, we then derive and discuss the partial susceptibilities in Equation 2 (Section 5), before we conclude (Section 6).

## 2   Simulations

We perform LES with the System for Atmospheric Modeling (SAM) (Khairoutdinov and Randall, 2003). Our domain measures $48 \times 48 \, \text{km}^2$ at a horizontal resolution of 200 m and a vertical resolution of 10 m. The timestep is 1 s. Simulations are nocturnal, and of 12 h duration. Sea surface temperature, subsidence and surface fluxes are fixed to the values of Ackerman et al. (2009). The model activates aerosol particles based on the prognosed supersaturation and simulates condensation/evaporation and collision-coalescence using a bin-emulating 2-moment approach (Feingold et al., 1998). Particles are removed by collision-

coalescence scavenging and wet deposition. We assume a surface source of aerosol particles of $70 \, \text{cm}^2 \, \text{s}^{-1}$ (Yamaguchi et al., 2017; Kazil et al., 2011). Integrated properties such as cloud water path (CWP), rain water path (RWP) and their sum, the LWP, are calculated directly from cloud and rain water mass mixing ratios. Drop number concentration is calculated from the prognosed cloud and rain number concentrations and is usually dominated strongly by the former. For more details on the model setup see Yamaguchi et al. (2017).

Following Feingold et al. (2016), we simulate 191 stratocumulus (Sc) cases with different initial conditions. We simultaneously vary six initial conditions of the Sc field: liquid water potential temperature ($284 < \theta_l/\text{K} < 294$), the total mixing ratio ($6.5 < q_t/(\text{g kg}^{-1}) < 10.5$) and the aerosol concentration ($30 < N_a/\text{cm}^{-3} < 500$) in a mixed layer, as well as the initial height ($500 < H_{\text{mix}}/\text{m} < 1300$) of that mixed layer, and the jumps ($6 < \Delta\theta_l/\text{K} < 10$, $-10 < \Delta q_t/(\text{g kg}^{-1}) < -6$) at the inversion above the mixed layer.

To generate our ensemble of model simulations, we sample well-spaced combinations of these initial conditions over the 6-dimensional space spanned by the given ranges using a maximin Latin-hypercube design algorithm (Morris and Mitchell, 1995). This Latin-hypercube sampling ensures good coverage across the 6-d space of the initial conditions and prevents any spurious aerosol-meteorology co-variability due to sampling (Feingold et al., 2016).





Out of a 200-point Latin-hypercube sampling, 191 initial conditions are identified that are expected to form cloud based on applying saturation adjustment to the initial conditions. For these initial conditions, actual LES are performed. From these simulations, we remove all those that do not sustain cloud. We also remove simulations that would form rain within the first hour while collision-coalescence and sedimentation are still switched off for spin-up. These simulations are characterized by

unrealistically strong rain once collisions-coalescence is allowed. We identify such early-precipitating simulations based on the criterion that the maximum domain-averaged surface precipitation rate over the timeseries $(0-12\,\mathrm{h})$ exceeds $10\,\mathrm{mm\,day^{-1}}$.

After this filtering, 159 of the initial 191 simulations remain. We discard two hours of spin-up and build our analysis on hours 2 to 12, at output intervals of $10\,\mathrm{min}$, which means that the timeseries from each simulation consists of 60 domain-averaged values. The total number of data points amounts to $60 \cdot 159 = 9540$.

Figure 1 summarizes our dataset as a function of the domain-averaged liquid water path LWP, which we define as the sum of cloud and rain water path, $\mathrm{LWP} = \mathrm{CWP} + \mathrm{RWP}$, and the vertically-averaged cloud droplet number concentration $N$ in cloud columns with $\tau(500\,\mathrm{nm}) > 1$. To illustrate the system evolution, we discuss the three labeled trajectories in the figure. The clouds of trajectory A deepen in response to longwave radiative cooling and attendant condensation. In contrast, the thick clouds of trajectory B feature strong entrainment that leads to cloud thinning. Cloud deepening and thinning approximately

balance each other in a region indicated by the solid blue line in Figure 1. Trajectory C shows a cloud system in which the adiabatic volume-mean droplet radius at cloud top reaches a critical value of about $12\,\mu\mathrm{m}$ associated with the onset of precipitation (Gerber, 1996). This rapidly reduces the cloud droplet number.

To guide further discussion, we partition the state space into four quadrants: The upper right quadrant ($1^{\mathrm{st}}$ quadrant Q1 in the following) is characterized by the absence of drizzle and evolution from the initial state towards decreasing LWP; the upper

left quadrant (Q2) features no drizzle and increasing LWP; the lower left quadrant (Q3) is barely sampled and characterized by drizzle and a diagonal evolution of increasing LWP and evolution towards decreasing $N$; the lower right quadrant (Q4) features drizzle and decreasing droplet number.

We distinguish the 2-dimensional *state space* spanned by LWP and $N$ from the *parameter space* of the system. Parameters are set externally and do not evolve in time. The parameters of our simulations are its boundary conditions, especially sea

surface temperature, subsidence and surface fluxes. Initial conditions could also be considered parameters. Their role for the system's evolution is, however, somewhat ill-defined due to the spin-up process. In real systems, a distinction between state variables and parameters is only approximate. It requires a timescale separation so that slowly evolving variables can be considered as fixed and parameter-like in comparison to fast evolving variables. While previous applications of emulators, e. g. Lee et al. (2013) and Johnson et al. (2015), have explored how the behavior of the system varies across the parameter

space, we explore how it varies across the resulting state space. Our choice of LWP and $N$ as state variables is motivated by Equation 2. It is not a priori clear that the properties of cloud fields that arise from a 6-dimensional set of initial conditions can be described as 2-dimensional functions. For such a reduction in dimensionality to be successful, it is required that multiple initial conditions in the 6-dimensional space map onto individual points on the 2-dimensional state space. In hydrology, this circumstance is known as *equifinality* (Beven, 2005). Our data does not perfectly, but only approximately, collapse onto the

2-dimensional state space. This imperfection manifests as noise on our data when presented in two dimensions.



**Figure 1.** Temporal evolution (hours 2-12) of 159 LES with varying initial conditions (see text) in an LWP-$N$ state space, colored by fraction of rain water path (RWP) to the total liquid water path (LWP). Individual simulations are indicated by gray lines and start at the location of gray circles. The dashed blue line corresponds to an adiabatic volume-mean droplet radius at cloud top of $12\,\mu$m (adiabatic condensation rate $\gamma = 2.5 \cdot 10^{-6}\,\mathrm{kg\,m^{-4}}$). Together with the solid blue line, it defines the quadrants labelled Q1 to Q4. Red letters indicate trajectories discussed in the main text.





## 3  Building ensembles of emulators

We analyze our data based on the assumption that the relationship between the state-space and the cloud output variables can be modeled as a Gaussian process, and employ the technique of *Gaussian process emulation* (Rasmussen and Williams, 2006). The historical origins of this method lie in predicting the distribution of gold in South Africa, based on a small sample

of carefully located test drillings. Gaussian process emulation is a preferred interpolation technique for sparse data (that is ideally well-spaced) of variables that vary smoothly (no discontinuities) across the dimensions of interest. Our data samples the LWP-$N$ state space partially in a sparse manner (when considering different runs) and partially in a reasonably dense manner (within the timeseries of individual runs). In contrast to the 6-dimensional initial conditions, which we sampled using a space-filling Latin hypercube design, the data associated with the system evolution is not Latin-hypercube sampled. As the

system evolves and moves towards the solid blue line in Figure 1, the coverage of the space becomes less evenly distributed, which can lead to issues of instability in parameter estimation when fitting a Gaussian process emulator. We therefore adapt the emulation approach to our situation. To fulfill the methodological requirements of sparsity and sampling, we generate a set of approximately Latin-hypercube sampled subsets of the data and construct a Gaussian process emulator for each subset (see Section 3.1). Together, this set of fitted emulators form an emulator ensemble that takes into account most of the available data

(see Section 3.2).

### 3.1  Sub-sampling

To build and validate the emulators in the emulator ensemble, we split the total dataset randomly into two equally large subsets. We use one of these to build the emulator (*training dataset*) and the other to independently test the emulator (*validation dataset*). The split is completely random and does not take into account to which timeseries a data point belongs.

Gaussian process-emulation is designed for reasonably sparse and well-spaced data over the dimensions of interest. As discussed, our data does not take this form because it consists of many densely-sampled timeseries that are themselves sparsely distributed in state space. We therefore sub-sample from the total training data in the following way: We create a virtual set of $n_{\mathrm{trn}}$ Latin-hypercube sampled points in the LWP-$N$ state space and replace each point in the Latin-hypercube sample by the geometrically closest point from our dataset. Data points are added when their normalized Euclidean distance from the

Latin-hypercube sample point does not exceed $5/n_{\mathrm{trn}}$. We do not use data points twice and training and validation data are treated as completely separate such that a data point cannot be selected for both training and validation. Figure 2 illustrates two such sub-samplings. We proceed in the same way to select a sub-sample of $n_{\mathrm{vld}} = 2n_{\mathrm{trn}}$ validation data points from the validation data set. The Latin-hypercubes underlying the samplings for the training and the validation dataset augment each other so that their combination is also Latin-hypercube sampled. We achieve the Latin-hypercube samples using the R package

`lhs` (R Core Team, 2018; Carnell, 2018).





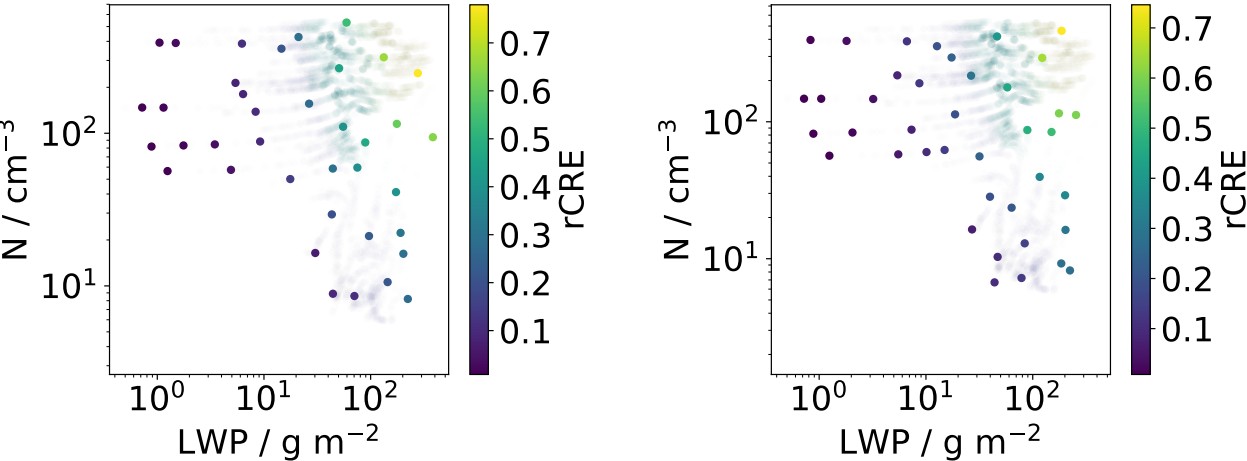

**Figure 2.** Sub-sampling (solid circles) from the total training data (opaque points) for the rCRE emulator ensemble members with the (left) lowest and (right) highest root mean square error (rmse) in predicting the validation data. The rmse values are 0.010 and 0.024, respectively.

### 3.2 Ensemble emulation and averaging

By varying the random seed of the initial Latin-hypercube sampling, we create an ensemble of sub-samplings. To each sub-sampling we apply Gaussian process emulation, constructing and validating a separate emulator model for each sub-sample in turn. For a general overview of the mathematical details of the emulator model, we refer to Johnson et al. (2015). It is

5  based on a Bayesian statistical framework, where we select an underlying mean and co-variance structure that is then fitted given information from the training data. We specifically assume a linear combination of LWP and N as the underlying mean function, and use a Matérn covariance structure (Rasmussen and Williams, 2006). We account for noise in our data (*nugget effect*). Our data is noisy because our simulations do not perfectly, but only approximately, collapse onto the LWP-$N$ state space. This is illustrated by the fact that individual data points in Figure 4 may differ in their value of the rCRE, i.e., their color,

10  from closely neighboring points. Emulators are fitted using the `km()` function in the R package `DiceKriging` (Roustant et al., 2012; R Core Team, 2018).

As the data points chosen for different Latin-hypercube samplings are not necessarily different, the emulator ensemble members that we obtain in this way are not independent. To limit the repeated use of data points, we relate the number of ensemble members $n_{\text{ensbl}}$, or random seeds, respectively, to the number of training data points for the individual emulators $n_{\text{trn}}$ such that no more than 50% of the total available training data $n_{\text{trn}}^{\text{tot}}$ will be used:

$$n_{\text{ensbl}} = \frac{n_{\text{trn}}^{\text{tot}}}{2 n_{\text{trn}}}.$$





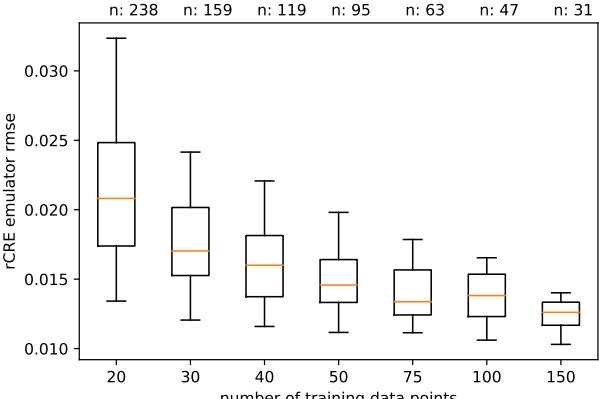

**Figure 3.** Standard deviation of individual rCRE emulator ensemble members obtained using different sub-samplings, as a function of the number of training data points. The number of ensemble members is indicated at the top of the plots. Orange lines, boxes and whiskers correspond to the median, upper and lower quartile, and 5th and 95th percentile of the distribution of standard deviations in the emulator ensemble.

We characterize an individual emulator within the emulator ensemble by its root mean square error (rmse) in predicting the validation data. We characterize the emulator ensemble as a whole by a weighted mean $\mu$, where the weighting $w$ depends on the rmse of the individual emulators:

$$\mu = \frac{\sum_i w_i x_i}{\sum_i w_i} \quad \text{with} \quad w_i = 1 - \frac{\text{rmse}_i - \min(\text{rmse})}{\max(\text{rmse}) - \min(\text{rmse})}. \tag{3}$$

5   For the example of the rCRE emulator shown in Figure 4, Figure 2 illustrates the best and worst sampling within the ensemble as quantified by the rmse of the corresponding individual emulators.

Figure 3 shows the spread of emulator rmse that are obtained when varying the number of training data points $n_{\text{trn}}$ used to build an emulator ensemble. The median rmse tends to decrease with an increasing number of data points, while the number of ensemble members decreases commensurately. As a compromise between the quality of individual emulators and ensemble

10   statistics we choose $n_{\text{trn}} = 50$.

## 4   Emulators for rCRE, cloud albedo and cloud fraction

Figures 4 and 5 demonstrate the successful application of our emulator ensemble technique to derive surfaces of rCRE, $A_c$ and CF (all determined using a threshold of $\tau > 1$) as a function of LWP and $N$ from the multi-timeseries data shown in Figure 1. The emulated surfaces successfully predict simulation outcomes (validation data) that were not used in creating the emulator

15   (training data) as shown by scatter plots and data points in the figures.

In accordance with Equation 1, the shape of the emulated rCRE surface follows from the surfaces of cloud albedo and cloud fraction (Figure 5, a,c). Cloud fraction generally decreases with decreasing LWP as the Sc deck entrains and thins until the



**Figure 4.** Emulated rCRE surface (ensemble mean Equation 3) as a function of LWP and droplet number $N$ as color contours. The standard deviation $\sigma$ of the emulated rCRE over the ensemble (Equation 4) is indicated by hatching, with $\sigma < 0.01$ and $0.01 < \sigma < 0.05$ for non-hatched and hatched regions, respectively. Emulated values outside $[0, 1]$ are masked. Color-filled circles show the validation subset of the total dataset shown in Figure 1 (see Section 3.1). For visibility only every 10th validation data point is shown. The scatter plots in the insets compare the complete validation dataset to the emulated values. White lines indicate quadrants as in Figure 1.

detection threshold $\tau = 1$ occurs. Its surface is dominated by a gradual shift of its isolines from Q2 to Q4 as rain formation sets in. In the shift region (Q3), cloud fraction depends strongly on $N$. Elsewhere, isolines run mostly in the vertical direction such that CF is largely independent of $N$. Cloud albedo is characterized by negatively-sloped isolines so that cloud albedo tends to



a) cloud fraction CF(LWP, $N$)    b) cloud fraction CF(CWP, $N$)

c) cloud albedo $A_C$(LWP, $N$)    d) cloud albedo $A_C$(CWP, $N$)

**Figure 5.** As Figure 4 but showing (a, b) cloud fraction and (c, d) cloud albedo as a function of (a, c) total liquid water path LWP and cloud droplet number concentration $N$ and (b, d) cloud water path CWP and $N$. The standard deviation $\sigma$ of the emulated rCRE over the ensemble (Equation 4) is indicated by hatching, with $\sigma < 0.01$, $0.01 < \sigma < 0.05$ and $0.05 < \sigma < 0.5$ for non-hatched, single-hatched, and double-hatched regions, respectively.

increase with both, LWP and $N$. The isolines, and thus the dependence of cloud albedo on LWP and $N$, are distorted in the drizzling region (Q3 and Q4).

This distortion in Q3 is caused by the bimodality of the drop-size distributions, so that cloud albedo and fraction are influenced by the radiative effects of cloud droplets as well as rain drops. Figure 5 (b, d) considers the cloud water path (CWP)




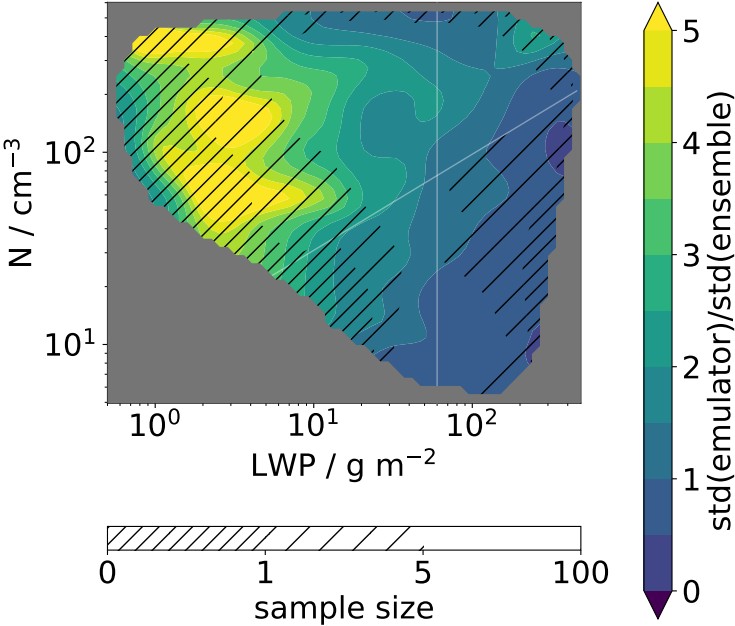

**Figure 6.** Comparison of errors from emulation, ensemble, and sampling. Contours show the ratio of emulator and ensemble standard deviation, hatchings show the sample size as in Figure 7. Note that sample sizes smaller than one trajectory can occur due to interpolation.

instead of LWP as x-axis. This transformation leads to a shift in the isolines: Cloud fraction isolines become approximately vertical as $\tau > 1$ is controlled by CWP alone and does not depend on the additional contribution of RWP to LWP. The tilt of the cloud albedo isolines is reversed, indicating a contribution of RWP to total cloud albedo.

## 4.1 Uncertainty

5    Our approach features five different kinds of uncertainties, or errors. First, the Gaussian process emulation returns a random variable, i.e. a probability distribution of possible surfaces. This uncertainty depends on the training data used to build an individual emulator. It can be quantified by a standard deviation and its values can be inferred from Figure 6.

     Second, the quality of the emulator mean function is quantified by the rmse of an emulator in predicting the validation data. This measure of uncertainty depends on the training as well as the validation data that is used for a specific emulator, i.e., one
10   member of the emulator ensemble. It is not spatially resolved but averages the error of the emulator over the whole LPW-$N$ state space. As indicated in the caption of Figure 2, rmse lies between 0.01 and 0.02 for the rCRE emulators.

     Third, we have the error due to the noise that arises when collapsing our dataset onto the LWP-$N$ state space. This uncertainty is the most fundamental uncertainty because it cannot be reduced by increasing the amount of data available. It is inherent to our modeling of rCRE as 2-dimensional function of LWP and $N$.





Fourth, we have the uncertainty of the emulator ensemble. Following Equation 3, we quantify this uncertainty using a weighted ensemble standard deviation,

$$\sigma = \sqrt{\frac{\sum_i w_i (x_i - \mu)^2}{\sum_i w_i}}, \tag{4}$$

where the weights $w_i$ are defined in Equation 3 and depend on the rmse. Hatchings in Figures 4 and 5 show that the ensemble

uncertainty (weighted standard deviation) for values of the rCRE is mostly smaller than 0.05 and in large regions of the state space smaller than 0.01. Comparing the different emulators, we find that the cloud-fraction emulator ensemble is more uncertain than that of the cloud albedo. This reflects that cloud albedo only depends on the local properties of the cloud field that are directly represented by LWP and $N$, while the cloud fraction is a cloud-field property. The cloud-fraction uncertainty is mitigated by considering the combined quantity of the rCRE.

Lastly, we have a sampling uncertainty illustrated in Figure 7. We indicate the level of sampling uncertainty by counting the number of trajectories that sample a specific part of the phase space. To this end, we partition the LWP-$N$ state space into $60 \times 50$ bins and for each bin we count how many of our 159 trajectories contain a data point within. This uncertainty arises because our data, especially when projected onto the 2-dimensional LWP-$N$ state-space, is noisy. Regions of the state space that are sampled by a single trajectory are thus less reliably represented than regions where the consideration of different

trajectories attenuates the noise. Note that the nugget effect assumed for building individual emulators cannot account for this sampling uncertainty: an undersampled region does not appear noisy.

Figure 6 compares the three spatially resolved types of error for the rCRE emulator. The standard deviation of the ensemble is mostly smaller than the standard deviation of the individual emulator. This reflects the larger amount of data and information considered when building the emulator ensemble. Comparison with the sampling uncertainty indicates that the ensemble

standard deviation may be overconfident in poorly sampled regions because it cannot capture the true level of noise in these regions. Therefore, because the ensemble uncertainty is generally small, we will use the sampling uncertainty to guide the interpretation of the rCRE emulator in the following.

### 4.2   Comparison to bilinear regression and effective degrees of freedom

Previous studies have determined partial susceptibilities as in Equation 2 from binned linear (Sena et al., 2016, e.g.,) or bilinear

regression (Jiang et al., 2010; Glassmeier and Lohmann, 2018). We therefore add a brief comparison of our method to bilinear regression. Figure 8 demonstrates that bilinear regression does not capture the simulation data as well as the emulator surface. While the coefficient of determination, $r^2 = 0.95$, and rmse$= 0.05$ are acceptable, the regression surfaces cannot account for the tilt of isolines discussed in Section 4. As a consequence, the regression surface predicts a large region of unphysically negative rCRE. The reason behind the poor performance of bilinear regression is that its 3 free parameters are insufficient

to capture the complexity of the emulated surface. The number of degrees of freedom required to adequately capture the complexity of the simulation data can be estimated from Figure 3. The value of $n_{trn}$ at which the emulator rmse levels off can be interpreted as the number of degrees of freedom of the fitted surface, in our case about 50. Performing bilinear regression for specific bins increases the total number of free parameters in the regression. In contrast to emulation, however, this approach



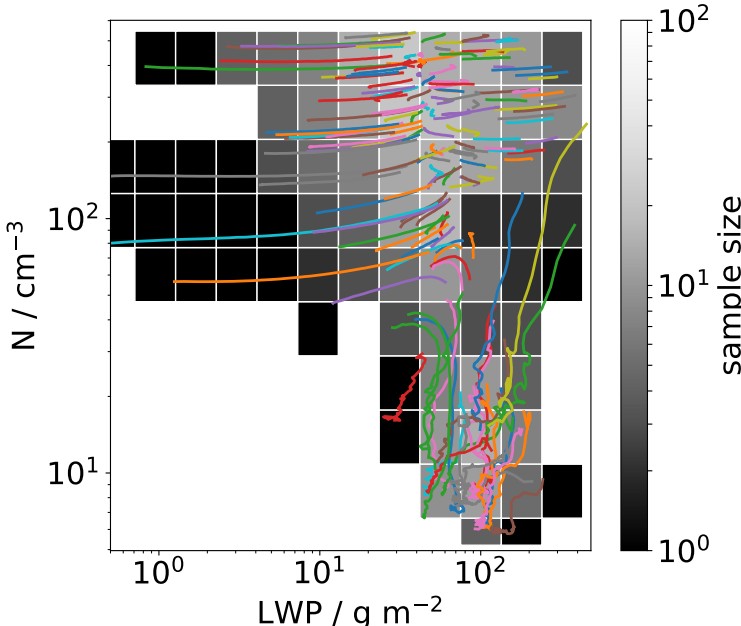

**Figure 7.** Illustration of sampling by trajectories. The sample size (greyscale) is determined by counting the number of trajectories (colored lines) per grid box. Note that the trajectories shown are the same as in Figure 1.

is limited by the requirement of a sufficient number of data points per bin. We therefore consider emulation to be a powerful and superior alternative to binned regression studies.

## 5   Partial susceptibilities of rCRE to droplet number and LWP

While partial susceptibilities (Equation 2) are directly obtained as the coefficients of a bilinear regression, the emulated rCRE

5   surface requires their derivation as finite differences of the array that represents the surface. Figure 9 shows the logarithmic derivatives of the surface shown in Figure 4.

In the upper part of the state space (Q1 and Q2), emulator-derived susceptibilities in Figure 9 (a, b) compare reasonably well to theoretical results for non-drizzling conditions (black line contours), which assume a unimodal droplet-size distribution and high-cloud fraction (Boers and Mitchell, 1994; Sena et al., 2016):

$$\frac{\mathrm{d}\ln \mathrm{rCRE}}{\mathrm{d}\ln \mathrm{LWP}} = \frac{5}{6}\left(1 - \mathrm{rCRE}\right), \qquad \frac{\mathrm{d}\ln \mathrm{rCRE}}{\mathrm{d}\ln N} = \frac{1}{3}\left(1 - \mathrm{rCRE}\right). \qquad (5)$$

In accordance with the different pre-factors, rCRE is thus more susceptible to LWP than to $N$. Susceptibilities decrease with increasing LWP, reflecting the saturation behavior of $A_{\mathrm{C}}$ for high LWP.





**Figure 8.** As Figure 4 but using bilinear regression $rCRE = -0.86 + 0.40 \cdot \log_{10}(LWP) + 0.26 \cdot \log_{10}(N)$ instead of emulation to obtain the surface. The coefficient of determination of the regression amounts to $r^2 = 0.95$ and the rmse$= 0.05$.

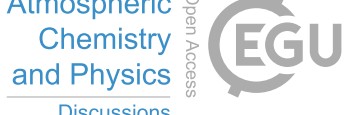



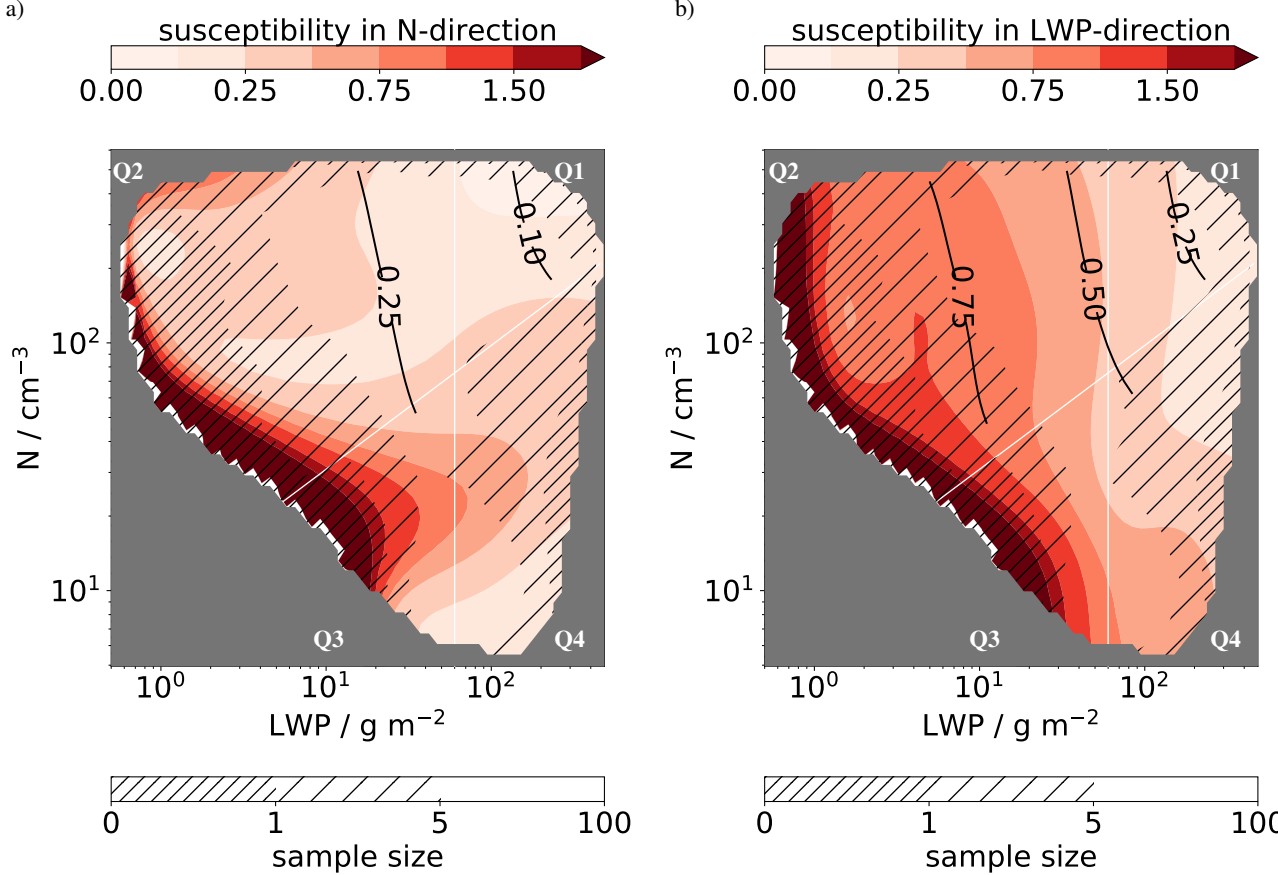

**Figure 9.** Partial susceptibilities of rCRE to (a) $N$ and (b) LWP as a function of $N$ and LWP as derived from the emulated rCRE surface (color contours). Solid lines show theoretical susceptibility values following Boers and Mitchell (1994), restricted to the non-drizzling region for which they apply. Hatchings indicate the sample size as in Figure 7. Note that sample sizes smaller than one trajectory can occur due to interpolation. White lines indicate quadrants as in Figure 1.

While Feingold et al. (1997) discuss the effect of drizzle initiation on $A_C$ and precipitation susceptibility, the authors are not aware of studies addressing susceptibilities of rCRE, $A_C$, CF or related quantities under drizzling conditions. Our emulator approach enables us to do so. As discussed in the context of Figure 5, the contribution of rain water to total LWP leads to a shift of rCRE isolines and makes cloud fraction at constant LWP a function of $N$. This creates a maximum in $\partial \ln \mathrm{rCRE}/\partial \ln N$ for

5   fixed LWP in the vicinity of the isoline distortion (Figure 9a). It also explains why isolines of $\partial \ln \mathrm{rCRE}/\partial \ln \mathrm{LWP}$ are tilted in the drizzling region. This indicates that the partial susceptibility of rCRE to $N$ not only captures the radiative effects of droplet size but also accounts for cloud fraction changes, while the partial susceptibility to LWP is comparably insensitive to the latter.

We abstain from interpreting the susceptibility in the left half of Q2; due to the low sample size in this region, the observed substructure is likely not physical.



## 6   Conclusions

We present a new method to summarize the detailed process representation ingrained in LES into a simple picture of aerosol-cloud interactions (Equation 2). We have constructed ensembles of Gaussian process emulators to extract how cloud albedo $A_\mathrm{C}$, cloud fraction CF, and the relative cloud radiative effect rCRE (Equation 1) depend on the domain-averaged liquid water path LWP and vertically-averaged cloud droplet number concentration $N$ from a set of 159 cloud-resolving simulations of nocturnal stratocumulus (Sc) with different initial conditions (Figure 1). The initial conditions were Latin-hypercube sampled from a 6-dimensional space that took into account variations of moisture and temperature profiles, including their jumps at the inversion, as well as aerosol concentration and boundary layer height. The emulator-ensemble approach has enabled us to accurately capture $A_\mathrm{C}$, CF and rCRE as a function of LWP and $N$ over a wide range of LWP and $N$ (Figures 4 and 5). Our results are based on an idealized set of simulations that currently do not account for varying boundary conditions like subsidence and surface fluxes. Taking such differences into account may lead to a broader and/or denser sampling of the LWP-$N$ state space. This would extend and improve the emulated surfaces.

Emulation provides a viable and more powerful alternative to multivariate linear regression for deriving cloud-state-dependent partial susceptibilities (Equation 2). We demonstrate this for the partial susceptibilities of rCRE to LWP and $N$ (Figure 9). We reproduce theoretical results for full cloud cover and monomodal droplet size distributions and extend the known relationships into the drizzling regime. As cloud fraction remains controlled by cloud water, the contribution of rain water to total LWP leads to a strong dependence of cloud fraction on $N$, for fixed LWP. This dependence corresponds to a strong susceptibility of rCRE to $N$ in the transition region from solid to broken cloud cover.

Our results confirm the expectation that rCRE is most susceptible to microphysical perturbations in the transition region between the high- and low-cloud fraction regime of Sc. Our new approach allows us to clarify the interpretation of Equation 2: The direct contribution of droplet number changes to rCRE, $\partial \ln \mathrm{rCRE} / \partial \ln N$, captures the effect of droplet number on cloud brightness as well as the effect on cloud fraction. This is possible because $N$ controls the rain water fraction RWP/LWP at constant LWP. The adjustment contribution, $\partial \ln \mathrm{rCRE} / \partial \ln \mathrm{LWP} \cdot \mathrm{d} \ln \mathrm{LWP} / \mathrm{d} \ln N$, captures the effect of cloud amount on rCRE, irrespective of its distribution onto fewer or more droplets, or thicker or thinner clouds at low or high cloud cover.

The methodology presented provides a powerful tool for synthesizing detailed data into a simple predictive framework. In this paper we have demonstrated its versatility for studying the sensitivities of cloud field properties over a wide range of states. Subsequent work will focus on employing this emulator approach to gain a deeper understanding of LWP adjustments $\mathrm{d} \ln \mathrm{LWP} / \mathrm{d} \ln N$. In general, computation-statistical approaches like the one discussed here have broad potential. They enable process modelers to explore their models beyond case studies, while at the same time they empower empiricists to better account for state dependence. This combination of approaches provides a promising avenue for improving our understanding of the uncertainties associated with the representation of shallow clouds in climate models.




*Code and data availability.* Input files and the model code for reproducing the simulation data of this study are available from the authors upon request.

*Author contributions.* FG carried out the analysis. All authors contributed to developing the basic ideas, discussing the results, and preparing the manuscript.

5   *Competing interests.* KC is executive editor, GF a co-editor of ACP. Other than this, the authors declare that they have no conflict of interests.

*Acknowledgements.* FG acknowledges support by a National Research Council Research Associateship award at the National Oceanic and Atmospheric Administration (NOAA) and by The Branco Weiss Fellowship – Society in Science, administered by the ETH Zürich. FH holds a visiting fellowship of the Cooperative Institute for Research in Environmental Sciences (CIRES) at the University of Colorado Boulder and the NOAA/Earth System Research Laboratory. TY is supported by the U.S. National Oceanic and Atmospheric Administration (NOAA)
10   Climate Program Office and by NOAA's Climate Goal. JJ and KC were supported by the Natural Environment Research Council (NERC) under grant NE/I020059/1 (ACID-PRUF) and the UK-China Research and Innovation Partnership Fund through the Met Office Climate Science for Service Partnership (CSSP) China as part of the Newton Fund. KC is currently a Royal Society Wolfson Merit Award holder. Marat Khairoutdinov graciously provided the SAM model.



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
