# Peer review of "An emulator approach to stratocumulus susceptibility"

_Atmospheric Chemistry and Physics, 2018_

## Referee Comment (RC1) · Anonymous Referee #1 · 19 Feb 2019

Overall Comment:

This work presents a strong foundation for modelling studies investigating the very complex nature of aerosol-cloud interactions. The methodology utilizes advanced computational tools that have been developed in recent years and applies them to this very challenging problem. The analysis corroborates the findings of existing literature demonstrating strong connections between cloud droplet number concentration and precipitation that are responsible for driving the liquid water path, cloud fraction and cloud radiative effect responses in aerosol-cloud interactions. The advantage of the emulator is evident in this context – it extends the analysis to a wider phase space than can be provided by sequential simulations alone. Overall, the paper is great, however,

[Figure]

I do have some concerns regarding the spatial and temporal scales of the training dataset simulations (described below) as it relates to the very important transitions between non-precipitating and precipitating clouds.

Minor Comments:

Pg3 L16: Spatial resolution: Is a 48 x 48 km domain big enough to simulate the largest scales of the boundary layer circulation? It is evident that this model can simulate a wide range of system-wide variables (e.g. cloud fraction) and (quite likely) mesoscale cloud types (e.g. closed vs open cells are probably being simulated here). The aspect ratios for mesoscale structures in stratocumulus can be quite large (e.g. 40:1; Wood and Hartmann, 2006) particularly for drizzling clouds. Drizzling stratocumulus require a larger LES domain in order to capture the interactions between precipitating clouds (Zhou et al. 2018, JAS). A concern, therefore, is if the emulator is constructed from poorly resolved mesoscale cloud interactions (for some of the simulations) then the inferred aerosol-cloud interaction responses will be distorted across the Latin-hypercube and the inferred responses for precipitating clouds that require a larger simulation domain would therefore bias the conclusions and interpretation of this study. Have the authors visualised the simulated cloud fields from these experiments? Adding a few representative images (possibly from each quadrant in figure 1) would be useful to the reader.

Pg 3 L16: Temporal resolution: Is 12 hours a long enough simulation period to observe transitions between non-precipitating and precipitating clouds? The reason I ask is because it is evident from the simulations shown in quadrant 1 of Figure 1 that very few of these simulations transition to a precipitating state? What is the reason for this? Could it be due to a short simulation period? Perhaps this point is mute because you have enough simulations to construct an emulator. Regardless, some justification for the chosen simulation period as it relates to this critical transition should be discussed in more detail.

Pg6 L1-15: Gaussian process emulation depends on stationary covariance functions and will perform poorly if the response surface has sharp local features, such as a discontinuity or a tall peak. I am wondering if this hypercube surface could be visualized from your analysis to both 1) convey how this rather esoteric concept works and 2) to boost confidence that Gaussian process emulation is a suitable assumption (as opposed to a non-stationary gaussian process emulation approach described here: Montagna and Tokdar, J. Uncertainty Quantification, 10.1137/141001512)

Figure 4: A recent paper (Rosenfeld et al. 2019, Science; 10.1126/science.aav0566) shows that increasing cloud droplet number concentrations increases both cloud top radiative cooling and precipitation suppression thereby causing cloud fraction to increase especially in larger thicker drizzling clouds. I would have therefore expected to see this effect in Figure 5b of your paper but the response appears somewhat flat. Why? Perhaps the differences between these papers are due to cloud diversity. For example, Rosenfeld sample a wide range of clouds between the equator and 40 South. I suspect that if the height of the inversion layer in this model is preventing the vertical development of deeper clouds (e.g. trade wind cumulus) and preventing transitions to precipitation this may be an explanation for why the relationship is dissimilar. This may also explain why the liquid water paths in this set of simulations is so much smaller than those observed from the satellite observations in Rosenfeld et al. 2019. Please discuss.

Other comments:

Pg1 L18: Warm shallow boundary layer clouds are also abundant and contribute substantially to the global aerosol indirect forcing-based estimate (e.g. Christensen et al. 2016, JGR; doi:10.1002/2016JD025245).

Pg4 L10: what is tau(500nm) >1? Is this the visible (at the 500 nm wavelength) optical depth? Please clarify.

Pg 6 L19: what does "completely random" mean?

Pg 7 L9 refers to Figure 4 but Figure 3 has not been discussed in the text yet (to this location).

Pg 7 L10: what is km () can you just spell it out?

Figure 4: "color-filled" circles; do you mean grey circles?

Pg 11 L1-3: CloudSat observations of the rainwater path contributions to total liquid water path have been shown to contribute significantly as the LWP becomes larger (Lebsock et al. 2011, JAMC, https://doi.org/10.1175/2010JAMC2494.1).

Figure 8 caption: Please include reference to where this equation was obtained.

---

## Referee Comment (RC2) · Anonymous Referee #2 · 5 Apr 2019

Review of An Emulator Approach to Stratocumulus Susceptibility

By Glassmeier, Hoffman, Johnson, Yamaguchi, Carslaw, and Feingold

Summary

This paper describes experiments in which an ensemble of large eddy simulations is used to train a statistical emulator. The purpose is to explore the sensitivity of the cloud radiative effect, as represented by cloud fraction and cloud albedo, to changes in droplet number concentration and liquid water path. Since it is not feasible to realistically perturb the LWP and N space directly, the authors instead perturb the initial and boundary conditions, then resample the resulting ensemble so that it more evenly spans the range of LWP and N. The emulator is then trained to reproduce the response

in Ac, CF, and rCRE as a function of changes in LWP and N.

The emulator appears to be able to effectively reproduce the behavior of Ac, CF, and rCRE over a range of LWP and N values. Perturbation of initial and boundary conditions is consistent with environmental controls on real Sc clouds, while the mapping to distribution of LWP and N allows for an effective exploration of the dependence of cloud radiative properties on cloud physical properties. The results indicate there are differences in response in drizzling and non-drizzling regimes, and that the emulator is more flexible in its representation of the cloud response than a linear regression methodology.

General Comments This paper presents an effective compromise between reductionist and emergent approaches, simplifying the former and adding more nuance and physical interpretation to the latter. The methodology is well described, and the sources of uncertainty are addressed. Overall, I find this to be a very interesting paper, and a nice contribution both scientifically and methodologically. The only issue I have is somewhat subtle and involves the partitioning of the LWP and N space into quadrants.

Specifically, I found the discussion of the four quadrants to be over-complicated - in particular, there does not appear to be much distinction between Q3 and Q4. The only plot that appears to show distinct behavior for Q3 vs Q4 is 5a, and the region of interest (the isolines of CF at lower left) is double-hatched, indicating much larger uncertainty in the emulator's ability to capture the behavior of the clouds in this region. Examination of Fig. 7 indicates there is perhaps only a single trajectory in this region of the state space, which makes me suspicious of the results. Instead of partitioning the drizzling portion of the state space into two distinct regions, I recommend first distinguishing between drizzling and non-drizzling (according to number) and then separating the non-drizzling cloud into those with large vs small LWP. This would result in three regions (Q1, Q2, and Q3+Q4), and I think the results based on the combined Q3+Q4 would be more robust. Certainly the key conclusion, that there is a strong dependence on N in the drizzling part of the state space, does not depend on dividing into Q3 vs Q4, right?

Specific Comments

1. p3, lines 30-31: Do the combinations of initial conditions make physical sense? E.g., do they correspond to physically realizable atmospheric states? LHS, as a space filling algorithm, does not necessarily respect the physical constraints known to be true of real environments, and often one must apply these constraints a posteriori to the LHS ensemble of initial conditions.

2. p12, line 4: Strictly speaking, the text here refers only to Fig. 4 not to Fig. 5, and I recommend removing the reference to Fig. 5 and instead making specific reference to it on line 7, which refers specifically to the cloud fraction results (shown in Fig. 5).

3. p12, line 10: I'm nit-picking here, but shouldn't the sampling uncertainty be Fig. 6 (not Fig. 7) and the combined uncertainty plot be Fig. 7 (not Fig. 6) so that they are in the proper order? In the current version, Fig. 7 is discussed before Fig. 6, which is a little strange.

---

## Author Comment (AC1) · 17 May 2019

We thank the reviewer for the careful and very detailed reading of the manuscript and for the encouraging and helpful comments. We always refer to the original, unrevised manuscript in the following.

**General comment**

This paper presents an effective compromise between reductionist and emergent approaches, simplifying the former and adding more nuance and physical interpretation to the latter. The methodology is well described, and the sources of uncertainty are addressed. Overall, I find this to be a very interesting paper, and a nice contribution both scientifically and methodologically. The only issue I have is somewhat subtle and

involves the partitioning of the LWP and N space into quadrants. Specifically, I found the discussion of the four quadrants to be over-complicated - in particular, there does not appear to be much distinction between Q3 and Q4. The only plot that appears to show distinct behavior for Q3 vs Q4 is 5a, and the region of interest (the isolines of CF at lower left) is double-hatched, indicating much larger uncertainty in the emulator's ability to capture the behavior of the clouds in this region. Examination of Fig. 7 indicates there is perhaps only a single trajectory in this region of the state space, which makes me suspicious of the results. Instead of partitioning the drizzling portion of the state space into two distinct regions, I recommend first distinguishing between drizzling and non-drizzling (according to number) and then separating the non-drizzling cloud into those with large vs small LWP. This would result in three regions (Q1, Q2, and Q3+Q4), and I think the results based on the combined Q3+Q4 would be more robust. Certainly the key conclusion, that there is a strong dependence on N in the drizzling part of the state space, does not depend on dividing into Q3 vs Q4, right?

*We agree with the referee's suggestion and have adapted text and figures accordingly.*

**Specific Comments**

1. p3, lines 30-31: Do the combinations of initial conditions make physical sense? E.g., do they correspond to physically realizable atmospheric states? LHS, as a space filling algorithm, does not necessarily respect the physical constraints known to be true of real environments, and often one must apply these constraints a posteriori to the LHS ensemble of initial conditions.

   *The referee is right that LHS sampling does not necessarily result in 'physically realizable' states. Specifically, not all combinations of initial conditions lead to cloud formation. As mentioned in p4, line 2f "Out of a 200-point Latin-hypercube sampling, 191 initial conditions are identified that are expected to form cloud based on applying saturation adjustment to the initial conditions." we indeed apply the cloud-forming constraint a posteriori. For clarification, we have replaced this*

*sentence by:*

"We create a Latin-hypercube sampling of 200 points. Not all of these correspond to conditions for which cloud formation is expected. Based on applying saturation adjustment as a posteriori condition, we identify 191 suitable initial conditions."

2. p12, line 4: Strictly speaking, the text here refers only to Fig. 4 not to Fig. 5, and I recommend removing the reference to Fig. 5 and instead making specific reference to it on line 7, which refers specifically to the cloud fraction results (shown in Fig. 5).

*This has been corrected accordingly.*

3. p12, line 10: I'm nit-picking here, but shouldn't the sampling uncertainty be Fig. 6 (not Fig. 7) and the combined uncertainty plot be Fig. 7 (not Fig. 6) so that they are in the proper order? In the current version, Fig. 7 is discussed before Fig. 6, which is a little strange.

*This has been improved accordingly.*

---

## Author Comment (AC2) · 17 May 2019

We thank the reviewer for the careful reading of the manuscript and for the encouraging and helpful comments. We always refer to the original, unrevised manuscript in the following.

**Minor comments**

1. Pg3 L16: Spatial resolution: Is a 48 x 48 km domain big enough to simulate the largest scales of the boundary layer circulation? It is evident that this model can simulate a wide range of system-wide variables (e.g. cloud fraction) and (quite likely) mesoscale cloud types (e.g. closed vs open cells are probably being simulated here). The aspect ratios for mesoscale structures in stratocumulus can

be quite large (e.g. 40:1; Wood and Hartmann, 2006) particularly for drizzling clouds. Drizzling stratocumulus require a larger LES domain in order to capture the interactions between precipitating clouds (Zhou et al. 2018, JAS). A concern, therefore, is if the emulator is constructed from poorly resolved mesoscale cloud interactions (for some of the simulations) then the inferred aerosol-cloud inter-action responses will be distorted across the Latin-hypercube and the inferred responses for precipitating clouds that require a larger simulation do- main would therefore bias the conclusions and interpretation of this study. Have the authors visualised the simulated cloud fields from these experiments? Adding a few rep-resentative images (possibly from each quadrant in figure 1) would be useful to the reader.

*We have added a figure (new figure 2) to visualize the simulated cloud fields and illustrate the spatial arrangement of the example trajectories A, B and C and also that of a new example trajectory D, to address minor comment 2. The manuscript on page 4, lines 12ff has been adapted accordingly:*

"To illustrate the system evolution, we discuss the four labeled trajectories in the figure. Figure 2 illustrates the spatial arrangement of these trajectories. The clouds of trajectory A deepen in response to longwave radiative cooling and at-tendant condensation. In contrast, the thick clouds of trajectory B feature strong entrainment that leads to cloud thinning. Cloud deepening and thinning approxi-mately balance each other in a region indicated by the solid blue line in Figure 1. Trajectory C shows a cloud system with large droplets whose adiabatic volume-mean droplet radius at cloud top quickly reaches a critical value of about $12\mu$m associated with the onset of precipitation (Gerber, 1996). This rapidly reduces the cloud droplet number (Figure 1) and leads to the break-up of the cloud field (Figure 2). Trajectory D initially features droplet sizes that are too small for pre-cipitation formation. Through cloud deepening similar to trajectory A, droplets grow until their size crosses the threshold value for precipitation formation. As for

trajectory C, this means that the cloud droplet number starts to decrease."

*The new figure shows that the drizzling trajectory C develops a mesoscale structure. We therefore capture mesoscale effects. As the referee points out, mesoscale structures in Sc are not expected to be much larger than 40 times the boundary layer height. In the drizzling region, our simulated boundary layers do not exceed depths of 1.4km. This means that we can captures mesoscale structures with aspect ratios up to 1:34. This lies within the typical aspect ratios of 1:30 to 1:40 given by Wood and Hartmann (2006).*

2. Pg 3 L16: Temporal resolution: Is 12 hours a long enough simulation period to observe transitions between non-precipitating and precipitating clouds? The reason I ask is because it is evident from the simulations shown in quadrant 1 of Figure 1 that very few of these simulations transition to a precipitating state? What is the reason for this? Could it be due to a short simulation period? Perhaps this point is mute because you have enough simulations to construct an emulator. Regardless, some justification for the chosen simulation period as it relates to this critical transition should be discussed in more detail.

*Precipitation formation can be considered a threshold process. When droplet sizes exceed a critical radius, 12μm in our case, precipitation forms within (tens of) minutes. On longer timescales, droplet sizes grow, when LWP increases or N decreases. Since the main mechanism to decrease N is precipitation formation itself, precipitation can only occur if LWP grows over time. This can be observed for several trajectories in region Q2 that start with comparably low droplet numbers, which are, however, initially not low enough to produce precipitation. As discussed in response to Minor Comment 1, we have added a new example trajectory D (see Figure 1 and new Figure 2), to explain this in the manuscript. For trajectories in region Q1, LWP tends to decrease so that droplets tend to shrink, rather than grow, which prevents precipitation formation and explains the lack of trajectories that slowly cross-over into the drizzling region Q34. We therefore do*

*not think that our simulation time is too short. Our simulation time of 12h was specifically motivated by the fact that we simulate a night-time situation, which would become unrealistic when run longer.*

3. Pg6 L1-15: Gaussian process emulation depends on stationary covariance functions and will perform poorly if the response surface has sharp local features, such as a discontinuity or a tall peak. I am wondering if this hypercube surface could be visualized from your analysis to both 1) convey how this rather esoteric concept works and 2) to boost confidence that Gaussian process emulation is a suitable assumption (as opposed to a non-stationary gaussian process emulation approach described here: Montagna and Tokdar, J. Uncertainty Quantification, 10.1137/141001512)

*We perform Gaussian process regression on two input variables (LWP and N) so that the response surface is 2-dimensional and can be visualized as contour plots (see Figures 4 and 5). Figure 4 illustrate the smoothness of our emulated rCRE surface. Together with the excellent skill of our rCRE emulator (see especially the inset of Figure 4), this makes us confident that Gaussian process emulation is applicable.*

4. Figure 4: A recent paper (Rosenfeld et al. 2019, Science; 10.1126/science.aav0566) shows that increasing cloud droplet number concentrations increases both cloud top radiative cooling and precipitation suppression thereby causing cloud fraction to in- crease especially in larger thicker drizzling clouds. I would have therefore expected to see this effect in Figure 5b of your paper but the response appears somewhat flat. Why?

*Since different RWP can occur for the same CWP, the effects of precipitation suppression are best studied in Figure 5a, where the contribution of RWP to LWP is taken into account. In the drizzling region Q34 of Figure 5a, an increase in cloud fraction with increasing N is clearly visible. In the non-drizzling regions Q1 and*

*Q2, Figures 5a and b are equivalent. As our CF emulators feature larger uncertainty than our rCRE emulators, the interpretation of a weak feature like a slight increase in cloud fraction with increasing N is difficult. Interestingly, the slope seems to increase when defining cloud fraction on $\tau > 5$, rather than $\tau > 1$ (see Figure 1 below). A detailed discussion of these observations is beyond the scope of the current paper, which focuses on rCRE and the emulator methodology, however.*

5. Perhaps the differences between these papers are due to cloud diversity. For example, Rosenfeld sample a wide range of clouds between the equator and 40 South. I suspect that if the height of the inversion layer in this model is preventing the vertical development of deeper clouds (e.g. trade wind cumulus) and preventing transitions to precipitation this may be an explanation for why the relationship is dissimilar. This may also explain why the liquid water paths in this set of simulations is so much smaller than those observed from the satellite observations in Rosenfeld et al. 2019. Please discuss.

   *In Figure 4 G, Rosenfeld et al. (2019) report median LWP values up to 160 g/m2. This is comparable with our values.*

**Other comments**

1. Pg1 L18: Warm shallow boundary layer clouds are also abundant and contribute substantially to the global aerosol indirect forcing-based estimate (e.g. Christensen et al. 2016, JGR; doi:10.1002/2016JD025245).

   *On page 1, line 18 we state, "the radiative forcing due to ACI, especially from shallow, warm clouds, continues to dominate the uncertainty margin of the total anthropogenic forcing". We clarify this by replacing:*

   "the radiative forcing due to ACI continues to dominate the uncertainty margin of the total anthropogenic forcing. Due to their abundance and location, forcing and

forcing uncertainty are dominated by shallow, warm clouds in marine boundary layers (Myhre et al., 2013; Boucher et al., 2013)."

2. Pg4 L10: what is tau(500nm) >1? Is this the visible (at the 500 nm wavelength) optical depth? Please clarify.

*We clarified this in the manuscript:*

"cloud columns with optical depth $\tau_{500\,\mathrm{nm}} > 1$, where the index indicates the considered wavelength."

3. Pg 6 L19: what does 'completely random' mean?

*We clarified in the manuscript:*

"The random splitting is based on equal, unconditioned probabilities for each data point to belong to either of the two datasets. It especially does not take into account to which time series a data point belongs and which data points from the same time series may already be in the same dataset."

4. Pg 7 L9 refers to Figure 4 but Figure 3 has not been discussed in the text yet (to this location).

*We improved the manuscript in the following way:*

"This is illustrated by the fact that individual data points in region Q34 of Figure 1 may differ in their value of RWP/LWP, i.e., their color, from closely neighboring points (cf. Figure 4 to see the same for rCRE instead of RWP/LWP)."

5. Pg 7 L10: what is km () can you just spell it out?

*km is the name of an r-function and presumably stands for 'Kriging model'. We have not found an official reference, however, that the authors of the function indeed chose to call their function 'km' as abbreviation for that term. We therefore only clarify:*

"Emulators are fitted using the function `km()` in the `R` package `DiceKriging`."

6. Figure 4: 'color-filled' circles; do you mean grey circles?

   *We clarified the figure caption like this:*

   "Color-filled circles with gray outline show the validation standard deviation of emulator ensemble subset of the total dataset shown in Figure 1 (see Section 3.1). For visibility only every 10th validation data point is shown. The colorscale for the data points is the same as for the contours and the gray outline of their edges has been added to distinguish validation data from the emulated surface (see Figure 8 for an example for this type of plot where values of data points and surface differ more)."

7. Pg 11 L1-3: CloudSat observations of the rainwater path contributions to total liquid water path have been shown to contribute significantly as the LWP becomes larger (Lebsock et al. 2011, JAMC, https://doi.org/10.1175/2010JAMC2494.1).

   *According to the reference, RWP contributes less than 5% to retrieved optical depth. We have accordingly specified the manuscript in the following way:*

   "Cloud fraction isolines become approximately vertical as $\tau > 1$ is controlled by CWP and hardly influenced by the additional contribution of RWP to LWP."

8. Figure 8 caption: Please include reference to where this equation was obtained.

   *We clarified:*

   "As Figure 4 but using bilinear regression $\mathrm{rCRE} = a \cdot \log_{10}(\mathrm{LWP}) + b \cdot \log_{10}(N) + c$ instead of emulation to obtain the surface. From the regression, we obtain $a = 0.40$, $b = 0.26$, $c = -0.86$ with a coefficient of determination of $r^2 = 0.95$ and rmse$= 0.05$."

**Fig. 1.** Supplementary plot for Figure 5 a for cloud fraction based on tau > 5 rather than tau >
1.